# The Histone Deacetylase Inhibitor ITF2357 (Givinostat) Targets Oncogenic BRAF in Melanoma Cells and Promotes a Switch from Pro-Survival Autophagy to Apoptosis

**DOI:** 10.3390/biomedicines10081994

**Published:** 2022-08-17

**Authors:** Adriana Celesia, Antonietta Notaro, Marzia Franzò, Marianna Lauricella, Antonella D’Anneo, Daniela Carlisi, Michela Giuliano, Sonia Emanuele

**Affiliations:** 1Department of Biomedicine, Neurosciences and Advanced Diagnostics (BIND), Biochemistry Building, University of Palermo, 90127 Palermo, Italy; 2Department of Biological, Chemical and Pharmaceutical Sciences and Technologies (STEBICEF), Laboratory of Biochemistry, University of Palermo, 90127 Palermo, Italy

**Keywords:** HDAC inhibitors, BRAF, melanoma cells, autophagy, apoptosis, epigenetic modifications

## Abstract

Histone deacetylase inhibitors (HDACI) are epigenetic compounds that have been widely considered very promising antitumor agents. Here, we focus on the effects of the pan-HDAC inhibitor ITF2357 (Givinostat) in comparison with SAHA (Vorinostat) in melanoma cells bearing BRAF V600E oncogenic mutation. Our results indicate both ITF2357 and SAHA dose-dependently reduce the viability of BRAF-mutated SK-MEL-28 and A375 melanoma cells. The comparison of IC50 values revealed that ITF2357 was much more effective than SAHA. Interestingly, both inhibitors markedly decreased oncogenic BRAF protein expression levels, ITF2357 being the most effective compound. Moreover, the BRAF decrease induced by ITF2357 was accompanied by a decrease in the level of phospho-ERK1/2. The inhibitor of upstream MEK activity, U0126, reduced ERK1/2 phosphorylation and dramatically potentiated the antitumor effect of ITF2357, exacerbating the reduction in the BRAF level. ITF2357 stimulated an early pro-survival autophagic response, which was followed by apoptosis, as indicated by apoptotic markers evaluation and the protective effects exerted by the pan-caspase inhibitor z-VADfmk. Overall, our data indicate for the first time that ITF2357 targets oncogenic BRAF in melanoma cells and induces a switch from autophagy to classic apoptosis, thus representing a possible candidate in melanoma targeted therapy.

## 1. Introduction

The proto-oncogene BRAF encodes a serine/threonine kinase, which acts upstream of the mitogen-activated protein kinase (MAPK) pathway, stimulating proliferative signals. Melanoma cells are frequently characterized by oncogenic BRAF mutations that cause the constitutive activation of the MAPK/ERK mitogenic pathway, resulting in increased cell proliferation [1,2]. The most well-known and recurrent BRAF mutation is the BRAF-V600E that results in a substitution of valine with glutamic acid. This mutation accounts for about 80% of the BRAF mutations in melanoma [3,4] and represents one of the major causes of melanoma transformation.

Melanomas bearing BRAF-V600E mutation tend to display peculiar clinical features and show a more aggressive behavior than BRAF wild-type melanomas [5]. In particular, evidence has been provided that BRAF-mutant tumors tend to metastasize more frequently and correlate with shorter overall survival in patients with advanced cancer than in those with BRAF wild-type melanoma.

The finding that so many melanomas harbor the BRAF-V600E mutation led to the development of the selective inhibitors of the oncogenic kinase, including Vemurafenib and Dabrafenib [6,7]. Although these inhibitors represent a significant advance in melanoma targeted therapy, unfortunately most patients often develop resistance to these drugs and relapse within a year, displaying the tendency of melanoma cells to acquire a resistant phenotype with increased aggressiveness [8,9,10].

From a molecular point of view, in addition to possible MAPK signaling activation by alternative mechanisms [11], melanoma cells can also develop resistance against Vemurafenib or Dabrafenib through the potentiation of pro-survival autophagy [12,13,14]. Autophagy represents a physiological adaptive response and a catabolic process that originates from the sequestration of damaged intracellular material into the autophagosome and consequent degradation following fusion with the lysosome. Autophagy is a multi-step process and requires the coordinated intervention of the ATG protein family [15,16]. Notably, many lines of evidence indicate that oncogenic BRAF can stimulate pro-survival autophagy in different tumor models [17,18,19] and that the autophagic process can exert a cytoprotective function against Vemurafenib-induced cell death in BRAF-mutated tumors, including melanoma [20,21,22,23].

Histone deacetylase inhibitors (HDACI) represent a class of epigenetic drugs with a remarkable anti-tumor activity and scarce toxicity in normal tissues [24,25]. By inhibiting histone deacetylates (HDACs), which frequently result overexpressed in many tumor forms, these compounds activate genes that were originally blocked by chromatin hypoacetylation. HDACs are thus considered important molecular targets to selectively affect tumor cells. HDACI have been also shown to specifically induce the expression of genes involved in cell differentiation and apoptosis [26]. HDACI and related molecules have also been recently shown to promote endoplasmic reticulum (ER) stress and consequent autophagy or apoptosis in different tumor models [27,28,29]. Intriguingly, some HDACI, including sodium butyrate and SAHA, have been shown to sensitize tumor cells bearing the BRAF-V600E mutation to the effect of vemurafenib [30,31].

Among HDACI, ITF2357 (Givinostat) is a pan-HDAC inhibitor that has been approved for the therapy of juvenile idiopathic arthritis and dystrophy muscular Duchenne [32] and displays significant anti-oncogenic potential [33]. The compound is currently in clinical phase studies for leukemia and lymphomas [34]. Some recent papers indicate that ITF2357 induces apoptosis in leukemic and glioblastoma cells [35]. Moreover, evidence has been provided that ITF2357 potentiates the effects of demethylating agents or chemotherapeutics, such as Pemetrexed, in lung cancer [36,37]. Another recent study demonstrates that ITF2357 induces apoptosis in sarcoma cells and sensitizes to the effects of doxorubicin [38]. However, although the effects of some HDACI in melanoma models have been described [39,40] so far there is no evidence in the literature for ITF2357 efficacy in melanoma cells.

This paper provides evidence that ITF2357 is particularly efficacious in inducing melanoma cell death and demonstrates for the first time that it is capable of targeting oncogenic BRAF and overcoming pro-survival autophagy, laying the foundations for translational studies and possible new melanoma-targeted therapy.

## 2. Materials and Methods

### 2.1. Chemicals and Reagents

ITF2357 (Givinostat) and SAHA (Vorinostat) were synthesized and kindly provided by Italfarmaco, Cinisello Balsamo, MI, Italy. For in vitro experiments, ITF2357 and SAHA were dissolved in DMSO (20 mM stock solution) and stored at −20 °C. Prior to use, stock solution was diluted in DMEM culture medium, not exceeding 0.01% (*v*/*v*) DMSO to realize the proper final concentrations. Equal volumes of DMSO were added to untreated cells as vehicle control.

The autophagy inhibitors (bafilomycin A1 and 3-methyladenine) and protein synthesis inhibitor cycloheximide were purchased from Sigma-Aldrich (Milan, Italy) and the general caspase-inhibitor z-VADfmk from Promega (Milan, Italy). Prior to use, stock solutions were opportunely diluted in DMEM culture medium, not exceeding 0.01% (*v*/*v*) DMSO, to realize the proper final concentrations. To study the MAPK pathway, we used the MEK inhibitor U0126 (Merck S.r.l., Milan, Italy), which was diluted in DMEM culture medium prior to use.

### 2.2. Cell Cultures

Human melanoma SK-MEL-28 (American Type Culture Collection, ATCC, Milan Italy) and A375 (gently provided by Prof. Mario Allegra, University of Palermo, Palermo, Italy) cell lines were grown in monolayer in 75 cm^2^ flasks in DMEM medium, supplemented with 10% (*v*/*v*) heat-inactivated fetal bovine serum (FBS), 2 mM L-glutamine, 100 U/mL penicillin, and 50 µg/mL streptomycin in a humidified atmosphere of 5% CO_2_ in air at 37 °C. For the experiments, cells were plated at a density of 5 × 10^3^ (SK-MEL-28) or 7 × 10^3^ (A375) in 96-well plates and at a density of 1.5 × 10^5^/well (SK-MEL-28) or 1.8 × 10^5^/well (A375) in 6-well plates, respectively, and allowed to adhere overnight. Subsequently, cells were treated with the chemicals or vehicle only and the incubation was protracted for the established times. All materials for cell cultures were purchased from Euroclone (Pero, Italy), and Life Technologies Ltd. (Monza, Italy).

### 2.3. Evaluation of Cell Viability

SK-MEL-28 and A375 cell viability was determined by 3-(4,5-dimethylthiazol-2-yl)- 2,5-diphenyltetrazolium bromide (MTT) assay. In brief, SK-MEL-28 and A375 melanoma cells were plated in 96-wells and treated with various concentrations of ITF2357 and SAHA for different times. After treatment, 20 µL MTT (Sigma-Aldrich, Milan, Italy) (5,5 mg/mL) were added and cells were incubated at 37 °C for 2 h. The medium was then removed, and cells were lysed with 100 µL of lysis buffer (20% sodium dodecyl sulfate in 50% N,N-dimethylformamide). Finally, the absorbance of the formazan was measured at 490 nm with 630 nm as a reference wavelength using an automatic ELISA plate reader (OPSYS MR, Dynex Technologies, Chantilly, VA, USA). Values reported in Figures are expressed as percentage of the viability of treated cells compared with vehicle treated (untreated control, 100% viability). The experiments were performed in triplicate and data are shown as mean ± SD of three independent experiments. IC50 were determined using the IC50 Calculator AAT Bioquest.

### 2.4. Evaluation of Autophagy by Monodansylcadaverine

Monodansylcadaverine (MDC) staining was performed to evaluate the formation of autophagic vacuoles as previously reported [41]. For these evaluations, SK-MEL-28 cells were plated in 96-well plates and treated with ITF2357. After 16 h treatment, cells were incubated with 50 µM MDC (Sigma Aldrich, Milan, Italy) for 10 min at 37 °C in the darkness. Cells were then washed with PBS and analyzed by fluorescence microscopy using a Leica DMR (Leica Microsystems, Milan, Italy) microscope equipped with a DAPI filter system (excitation wavelength of 372 nm and emission wavelength of 456 nm). Images were acquired by computer imaging system (Leica DC300F camera, Milan, Italy). Three different visual fields were examined for each condition.

### 2.5. Detection of Chromatin Condensation by Hoechst Staining 

Cell death was assessed by staining the cells with the vital dye Hoechst 33342 (Sigma-Aldrich, Milan, Italy), which shows nuclei and allows for the detection of chromatin condensation and fragmentation. For these experiments, 7 × 10^3^ cells/well were seeded in 96-well plates, incubated with the compounds for the established times and then stained with Hoechst (2.5 µg/mL medium) in the dark for 30 min. After washing with PBS, cells were visualized using an inverted Leica fluorescent microscope (Leica Microsystems, Wetzlar, Germany) with a 4′,6-diamidino-2-phenylindole dihydrochloride (DAPI) filter (excitation wavelength of 372 nm and emission wavelength of 456 nm). Images were acquired through a computer imaging system (Leica DC300F camera, Milan, Italy). Three different visual fields were examined for each condition.

### 2.6. Western Blot Analysis

For Western blot analysis, whole-cell extracts were prepared in ice-cold lysis RIPA buffer (1% NP-40, 0.5% sodium deoxycholate and 0.1% SDS in PBS, pH 7.4), supplemented with a protease inhibitor cocktail, and subjected to SDS PAGE and consequent immunoblot as previously reported [42]. In these experiments, the correct protein loading was verified by both Ponceau red staining and housekeeping protein γ-tubulin immunodetection. Specific primary antibodies directed against BRAF, ERK1/2, p-ERK1/2, and caspase 3 (diluted 1:500), were purchased from Santa Cruz Biotechonology (St.Cruz, CA, USA); γ-tubulin, LC3, and p62 (diluted 1:1000) from Sigma-Aldrich (Milan, Italy); and ATG7, Beclin, caspase 9, and PARP-1 (diluted 1:1000) from Cell Signaling Technology (Beverly, MA, USA). Immunodetection was carried out by electrochemical luminescence labelling system (ECL) using ChemiDoc, XR Image system (Bio-Rad Laboratories, Hercules, CA, USA). The intensity of the protein bands was quantified using Quantity One Imaging Software (BioRad Laboratories) and reported as the ratio of the intensity of protein bands normalized to γ-tubulin, versus the intensity of the untreated samples, if not differently indicated. All the blots shown in Figures are representative of three independent experiments.

### 2.7. Semiquantitative RT-PCR

RNA was extracted by Direct Zol RNA Mini-Prep (Zymo research, Freiburg, Germany). A DNase I treatment step was included. One microgram of total RNA was reverse-transcripted in a final volume of 20 µL by using QuantiTect^®^ Reverse Transcription Kit (Qiagen, Germany). The resulting cDNAs were used for semi-quantitative PCR (RT PCR) by using Euro Taq thermostable DNA polymerase kit (Euroclone, Mi, Italy) according to the manufacturer’s instructions. Primers against BRAF were the following: BRAF-51F (forward 5′-CTACTGTTTTCCTTTACTTACTACACCTCAGA-3′ and BRAF-176R reverse 5′-ATCCAGACAACTGTTCAAACTGATG-3′). The cycling protocol used was as follows: initial denaturation at 95 °C for 5 min, followed by denaturation at 95 °C for 30 s, annealing at 62 °C for 50 sec, extension at 72 °C for 30 s for 35 cycles, and a final extension at 72 °C for 5 min. Primers against rRNA18S were used as control to demonstrate the equal loading of RNA (initial denaturation at 95 °C for 3 min, denaturation at 95 °C for 1 min, annealing at 55 °C for 1 min, extension at 72 °C for 1 min for 25 cycles, and a final extension at 72 °C for 10 min). The amplified products were resolved by agarose gel electrophoresis (1% agarose, 0.5 µg/mL ethidium bromide; Sigma-Aldrich, Milan, Italy), and the bands were visualized and photographed with ChemiDoc XRS (Bio-Rad Laboratories Srl, Milan, Italy). Data processing and densitometric analysis were performed by using Quantity One Analysis Software from Bio-Rad Laboratories.

### 2.8. Statistical Analysis 

Data were represented as mean ± S.D., and analysis was performed using the Student’s *t*-test and one-way analysis of variance. Comparisons between untreated control vs. all treated samples were made. If a significant difference was detected by ANOVA analysis, this was re-evaluated by post hoc Bonferroni’s test. GraphPadPrismTM 4.0 Software (Graph PadPrismTM Software Inc., San Diego, CA, USA) was used for statistical calculations. The statistical significance threshold was fixed at *p* < 0.05.

## 3. Results 

### 3.1. ITF2357 Potently Reduces Melanoma Cell Viability and Induces Histone Acetylation

To date there are no data in the literature on the effects of ITF2357 (Givinostat) in melanoma models, while other HDAC inhibitors have been tested showing anti-tumor efficacy [31,43]. Therefore, we focused on the effects of ITF2357 on two BRAF V600E mutated cell lines, SK-MEL-28 and A375 cells, to see whether this epigenetic compound could represent a promising therapeutic candidate. Initially, the effects of different doses of ITF2357 were evaluated on melanoma cell viability by MTT assay in comparison with the well-known HDAC inhibitor suberoylanilide hydroxamic acid (SAHA, vorinostat). As shown in Figure 1, after 48 h treatment, both ITF2357 and SAHA reduced the viability of SK-MEL-28 and A375 cells in a dose-dependent manner. 

It is interesting to note that ITF2357 resulted much more efficacious than SAHA in both cell lines as revealed by the hystograms and comparison of IC50 values reported in Table 1. Moreover, these evaluations revealed that A375 cells were more susceptible to both HDAC inhibitors.

These results indicate that ITF2357 efficaciously exerts cytotoxic effects in melanoma cells and can thus be considered an HDAC inhibitor with a particular anti-tumor efficacy.

Given that the inhibitory effects of SAHA on HDAC activity have been widely described [44,45], in order to confirm that ITF2357 also behaves as an HDAC inhibitor in our model, we verified that it was capable of inducing histone acetylation. Western blot analysis reported in Figure 2 shows that the compound markedly induced an increase in the levels of acetylated H3 and H4 histones in both SK-MEL-28 and A375 cell lines. This effect was visible at 16 h treatment and maintained at 48 h, thus suggesting that ITF2357 is also capable of promoting HDAC inhibition in both melanoma cell lines. 

### 3.2. The Effects of ITF2357 on Oncogenic BRAF 

Considering the key role of mutated V600E BRAF in promoting melanoma growth and aggressiveness [3], we wondered whether ITF2357 could target this protein or modify its expression levels. To address this point, Western blot analysis was performed using both ITF2357 and SAHA as HDAC inhibitors (Figure 3). Interestingly, both compounds reduced the level of oncogenic BRAF in a dose-dependent manner in both melanoma cell lines. In SK-MEL-28 cells, the reducing effect was already remarkable with 2 μM ITF2357 after 48 h treatment, while SAHA exerted a similar reduction in BRAF level with a dose of 40 μM, thus confirming the much higher efficacy of ITF2357 compared to SAHA. Notably, the BRAF band almost disappeared with 5 μM ITF2357 and was not visible at all with 10 μM. In A375 cells, which were more sensitive to both HDAC inhibitors, the reducing effects were remarkable even with 1 μM ITF2357. As regards the effects of SAHA, BRAF decrease was visible with a much higher concentration (from 10 μM to 40 μM), thus confirming the higher efficacy of ITF2357. These data indicate for the first time that both the HDAC inhibitors target oncogenic BRAF and that ITF2357 is the most efficacious compound.

We thus concentrated on ITF2357 using the concentrations near to the IC50 for each cell line and tried to elucidate the reason for BRAF decrease in treated cells. First, since ITF2357 is an epigenetic compound, we considered a possible effect on BRAF gene expression. As shown in Figure 4a, RT-PCR indicated that ITF2357 also determined a reduction in BRAF mRNA in both cell lines. This was evidenced at 24 h treatment, a time that was considered proper since the cells were still viable and degradation processes had not occurred yet. However, since the BRAF protein decrease was observed at 48 h and was dramatic, we also considered the possibility that protein degradation events could also contribute during this second phase of treatment in addition to reduced expression. Considering that BRAF can be degraded by the 26S proteasome [46], we investigated whether the proteasome inhibitor bortezomib could attenuate the reducing effects of ITF2357. Bortezomib is another antitumor agent that has been widely described in the literature [47,48]. For combination experiments, we chose a concentration that was not toxic to melanoma cells but that was reported to inhibit the proteasome [49]. As shown in Figure 4b, the addition of bortezomib to ITF2357-treated cells in the second phase of treatment (last 24 h) consistently reduced the BRAF decrease. These data suggest that proteasome-mediated degradation might also account for the remarkable BRAF protein level decrease observed with the HDAC inhibitor. To confirm the induction of degradation, we compared the half-life of the BRAF protein in the presence and absence of ITF2357. The results shown in Figure 4c indicate that ITF2357 reduced the half-life of BRAF, anticipating the decreasing effect under protein synthesis inhibition by cycloheximide. 

### 3.3. The Effect of ITF2357 on BRAF Mitogenic Signalling Cascade

To understand whether BRAF-targeting by ITF2357 was correlated with a reduction of BRAF-mediated mitogen activated kinase (MAPK) signaling, we evaluated the levels of phospho-ERK1/2, the downstream kinase in this pathway. Western blot analysis performed at different treatment times showed that the level of phospho-ERK1/2 was not significantly modified after 16 h treatment with ITF2357, although it decreased at 48 h similarly to the level of BRAF (Figure 5a). It is thus possible to speculate that the decrease in BRAF, which is quite a late event, accounts for the reduced activation of the MAPK mitogenic signaling. In addition, as a further confirmation, we evaluated the effects of U0126, which inhibits MEK, the kinase that is directly activated by BRAF and consequently promotes downstream ERK1/2 phosphorylation. As shown in the same Figure 5, U0126, which produced a cytostatic effect when used alone, markedly potentiated the effect of ITF2357 on both cell viability (panel b) and cell morphology (panel c). It is interesting to note that U0126 completely suppressed ERK phosphorylation and exacerbated the decreasing effect of ITF2357 on BRAF levels, as revealed by Western blot analysis (panel d). 

### 3.4. ITF2357 Promotes a Switch from Autophagy to Caspase-Dependent Apoptosis

A number of papers have correlated the oncogenic BRAF function with the promotion of pro-survival autophagy and consequent melanoma cell survival and propagation [14,50,51].

In order to understand whether ITF2357 induces autophagy in our model, we first used monodansylcadaverine (MDC) staining to detect autophagosome formation. As shown in Figure 6 (upper panel), the basal autophagy level was detected in SK-MEL-28 cells as revealed by green fluorescence, an effect that markedly increased following ITF2357 treatment for 16 h, where brilliant dot-like structures were clearly visible. Prolonging treatment time up to 48 h revealed a dramatic reduction in green fluorescence intensity, whereas signs of chromatin condensation and fragmentation appeared as evidenced by nuclei staining with Hoechst (lower panel). Similar results were obtained in A375 cells (not shown).

To confirm the morphological evidence of autophagy and/or apoptosis induction, we evaluated the effects of ITF2357 on autophagic and apoptotic markers at two time points (16 and 48 h) in both melanoma cell lines. As shown in Figure 7, ITF2357 induced the production of LC3-II from LC3-I, an effect that was particularly evident in A375 cells already at 16 h and increased at 48 h, indicative of autophagosome formation. 

P62 is a well-known multifunctional protein involved in selective autophagy and usually studied to monitor the autophagic flux [52]. The levels of p62 generally increase early upon the stimulation of the process and then decrease when autophagy is completed since the protein becomes degraded by autophagolysosomes. We found that ITF2357 induced a significant increase in p62 level, which was further enhanced at 48 h in both cell lines, most likely due to the accumulation of the protein and the lack of autophagy completion. We also evaluated the levels of beclin and Atg7, two other autophagic markers, which tended to decrease at 48 h. Such effects were more pronounced in A375 cells that were more sensitive to ITF2357 than SK-MEL-28 cells.

We can interpret these data considering that the triggering of the autophagic flux most likely represents an adaptive cell response to the effects of compound. To better understand whether ITF2357-induced autophagy is activated as a pro-survival mechanism, we evaluated cell viability in the presence of bafilomycin A1 and 3-methyladenine, two autophagy inhibitors that act at different levels of autophagy execution [53]. Data reported in Figure 8 show that the effect of ITF2357 was significantly potentiated by both autophagy inhibitors, thus suggesting that autophagy has a predominant pro-survival meaning. 

The same figure shows the effects of the general caspase inhibitor z-VADfmk, which almost completely prevented the effect of ITF2357. This result supports that caspase-dependent apoptosis accounts for ITF2357-induced cell death. 

The involvement of apoptosis was confirmed by the evaluation of apoptotic markers, caspase-9, caspase-3, and the caspase substrate poly-ADP-ribose polymerase-1 (PARP) in both cell lines. It is interesting to note that caspase activation occurred at 48 h, as evidenced by the remarkable decrease in pro-caspase 3 and the appearance of the active fragment of caspase-9 (Figure 9). These effects were accompanied by PARP cleavage, detected by the 85 kDa fragment display, thus confirming that ITF2357 activates a caspase-dependent apoptosis in the second phase of treatment.

## 4. Discussion

Our data demonstrate for the first time that HDACI (ITF2357 and SAHA) target oncogenic BRAF in melanoma cells. These findings provide a rationale to explain the effects previously observed by López-Cobo et al. who found that HDACI sensitize tumor cells bearing the BRAF-V600E mutation to the effect of BRAF inhibitors [30]. The decreasing effect on the oncogenic BRAF protein level was particularly evident with ITF2357, which resulted the most efficacious HDAC inhibitor in SK-MEL-28 and A375 melanoma cells. For this reason, the manuscript was focused on ITF2357 as a promising novel HDAC inhibitor in melanoma cells. Specifically, the concentrations used in SK-MEL-28 and A375 cells were chosen in accordance with the respective IC50 values. 

This compound was also capable of inducing histone acetylation in both cell lines, a result in accordance with the findings of Del Bufalo et al., who found histone acetylation by ITF2357 in another cancer model [37].

It is not clear yet whether histone acetylation can contribute to BRAF decrease. However, we can hypothesize that epigenetic changes induced by the HDACI-modifying gene expression pattern reduces BRAF gene transcription. This hypothesis is sustained by our data that indicate that BRAF mRNA levels decrease following ITF2357 treatment in both cell lines. 

Although histone acetylation usually favors chromatin relaxation and consequent gene expression, it should not be excluded that, possibly, overall epigenetic changes due to the histone code may result in repressing specific genes. Regarding this perspective, Kaimori et al. have shown that the acetylation of specific histone tail residues together with other histone modifications can promote gene repression [54]. It is thus not surprising that epigenetic changes may result in reduced BRAF expression. 

We also provided evidence that proteasome-mediated degradation can also account for the remarkable BRAF protein level decrease observed with the HDAC inhibitor, since the proteasome inhibitor bortezomib partially prevented the decreasing effect of ITF2357 on BRAF levels. Accordingly, Hernandez et al. have clearly demonstrated that BRAF is degraded by the 26S proteasome, following interactions with FBXW7 ubiquitin ligase [55]. 

The evaluation of the BRAF half-life using cycloheximide confirmed that BRAF undergoes degradation under the effect of ITF2357 since the compound anticipated the decrease observed in the condition of protein synthesis inhibition. It is interesting to note that in normal conditions the BRAF decrease induced by ITF2357 was quite a late event (observed at 48 h), similar to the decrease in the downstream signaling component, phospho-ERK. The decreasing effect of ITF2357 that we observed on phospho-ERK confirmed that the BRAF-targeting by the compound was accompanied by inhibition of the mitogenic MAPK pathway. Moreover, we showed that the MEK inhibitor U0126 markedly potentiated the effects of the HDAC inhibitor on both cell viability and BRAF levels. 

These data suggest that combining the HDAC inhibitor with the MEK inhibitor increases oncogenic BRAF-targeting and may represent a nice tool to potentiate melanoma cell death. In line with our findings, recent evidence has been provided that HDACI can enhance the antitumor activity of MEK inhibitors in other tumor models [56]. Concerning the effect of ITF2357 in the presence of U0126 on BRAF protein level, it is possible to speculate that the MAPK mitogenic signaling exerts a positive feedback loop on BRAF expression. In this regard, a fascinating hypothesis has been proposed by Saei et al., who demonstrated that the deubiquitinating enzyme USP28, which destabilizes BRAF, is regulated through a feedback loop promoted by the same BRAF mitogenic signaling [57]. Specifically, in this loop the inhibition of USP28 favors BRAF stabilization. The inhibition of the mitogenic signaling with the consequent increase in USP28 could thus promote BRAF destabilization. This hypothesis is in line with our observation that the BRAF decrease observed with ITF2357 depends not only on reduced expression but also by degradative processes as well. In this regard, ongoing studies in our laboratory aim to confirm that BRAF is degraded by the 26S proteasome in melanoma cells.

Much evidence has been provided that pro-survival autophagy is promoted by oncogenic BRAF, thus favoring melanoma cell survival and propagation [14,50,51]. More generally, oncogenic BRAF mutations have been associated with autophagy-promoted tumorigenesis and tumor progression in other tumor models [17,18,19]. Interestingly, Angeletti et al. recently provided evidence that autophagy inhibition strongly potentiates the efficacy of ITF2357 in human glioblastoma cancer stem cells [58].

Here, we showed that ITF2357 induces a pro-survival autophagic response since the autophagy inhibitors bafilomycin A1 and 3-methyladenine markedly increased the cytotoxic effect of the HDAC inhibitor. The evaluation of autophagic markers at different time points confirmed this trend since most of them increased early (16 h) and tended to decrease later on (48 h), with the exception of p62, which accumulated at 48 h, suggesting that autophagy was not completed. Interestingly, caspase-dependent apoptosis was detected at 48 h and confirmed by the protective effect of the pan-caspase inhibitor z-VADfmk. Based on these results, we hypothesize that ITF2357 promotes a switch from autophagy to classic apoptosis. The finding that ITF2357 induces apoptosis is in line with the observations of other authors who described the pro-apoptotic effect of the compound in other tumor cell lines [33,38,59].

## 5. Conclusions

Overall, our data indicate that the HDAC inhibitor ITF2357 has a potential as melanoma therapy candidate since it targets oncogenic BRAF protein. This effect is most likely due in part to reduced BRAF expression and in part to BRAF protein degradation. We also provided evidence that BRAF mitogenic signaling is reduced in the presence of ITF2357 and that MEK inhibition dramatically potentiates the effects of the compound. According to this finding, the pharmacological inhibition of the signaling may be proposed in association with ITF2357 to strategically kill melanoma cells. We also showed that ITF2357 promotes a switch from pro-survival autophagy to caspase-dependent apoptosis and, according to our results, we suggest that autophagy inhibition may also represent a tool to improve the compound efficacy. 

## Figures and Tables

**Figure 1 biomedicines-10-01994-f001:**
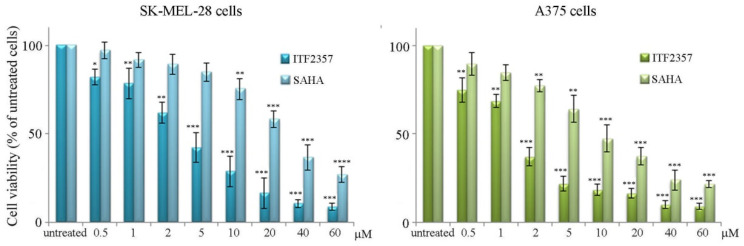
The effects of ITF2357 and SAHA on melanoma cell viability. For the evaluation of cell viability, SK-MEL-28 and A375 cells were treated with the indicated concentrations of ITF2357 or SAHA for 48 h. MTT analysis was then carried out as reported in Materials and Methods. The results reported in the histograms are representative of three independent experiments. * *p* < 0.05, ** *p* < 0.01, *** *p* < 0.001, **** *p* < 0.0001 with respect to untreated cells.

**Figure 2 biomedicines-10-01994-f002:**
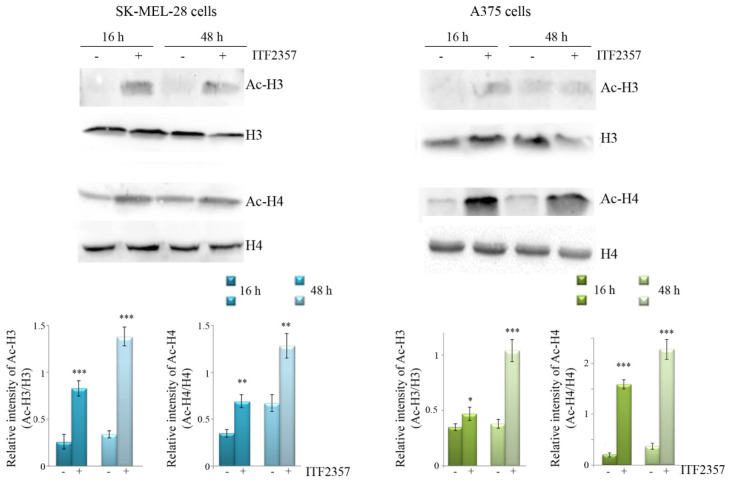
ITF2357 promotes histone acetylation in melanoma cells. Western blot analysis of acetylated H3 and H4 histones after treatment for 16 and 48 h with 5 μM (SK-MEL-28) or 2 μM (A375) ITF2357. The ratio between acetylated and total histone levels was quantified. Representative blots of three independent experiments and densitometric analysis are shown. * *p* < 0.05, ** *p* < 0.01, *** *p* < 0.001 with respect to untreated cells.

**Figure 3 biomedicines-10-01994-f003:**
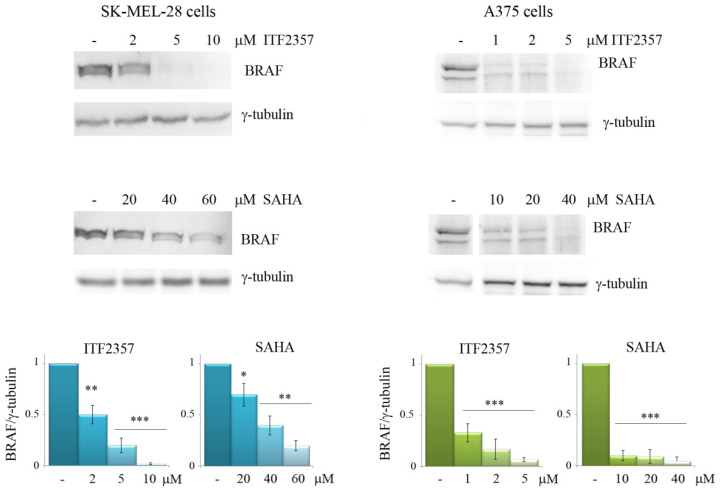
ITF2357 and SAHA dose-dependently decrease oncogenic BRAF levels. SK-MEL-28 and A375 cells were treated with the indicated concentrations of the two HDACI for 48 h. Western blot analysis of BRAF was performed as reported in Materials and Methods. Representative blots of three independent experiments and densitometric analysis are shown. * *p* < 0.05, ** *p* < 0.01, *** *p* < 0.001 with respect to untreated cells.

**Figure 4 biomedicines-10-01994-f004:**
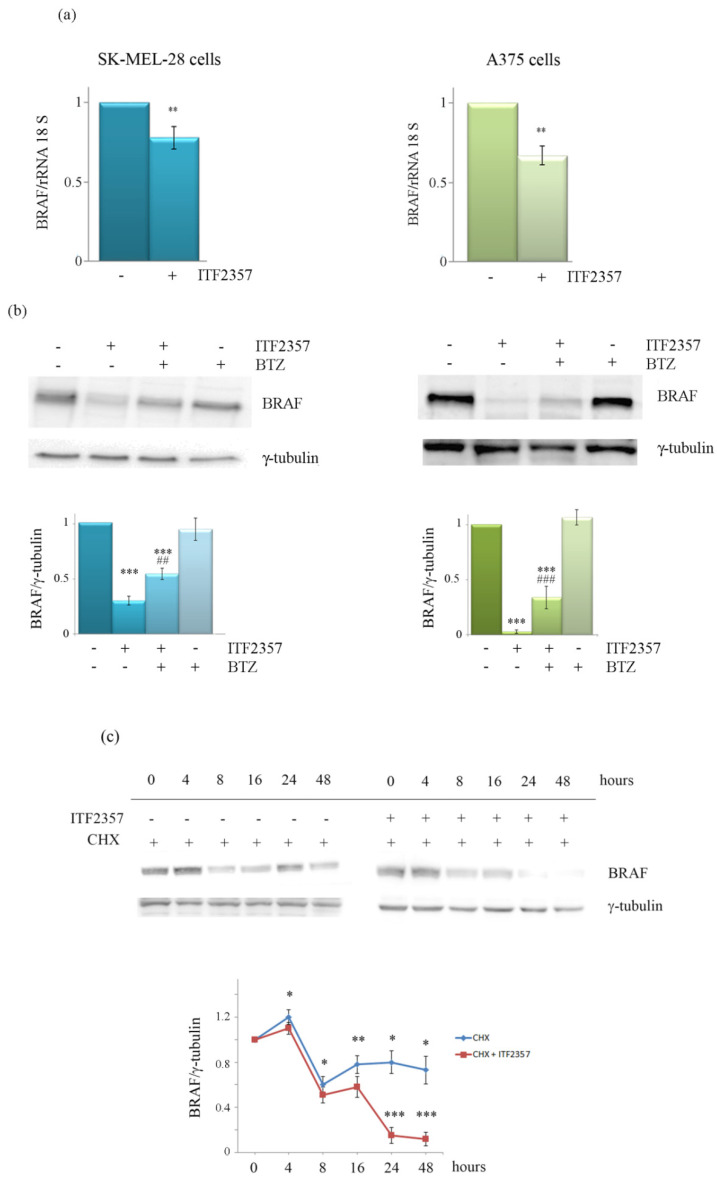
The effects of ITF2357 on BRAF expression (**a**) and protein degradation (**b**,**c**). SK-MEL-28 and A375 cells were treated for 24 h with 5 μM and 2 μM ITF2357, respectively. RNA extraction and RT-PCR were then performed as reported in Materials and Methods. Histograms are representative of two independent experiments. (**b**) The effect of proteasome inhibition by bortezomib (BTZ) on the decreasing effect of ITF2357. SK-MEL-28 and A375 cells were treated for 24 h with 5 μM and 2 μM ITF2357 respectively, then BTZ (10 nM) was added, and the incubation was protracted for another 24 h. (**c**) BRAF half-life was determined using 100 μM cycloeximide for the indicated times in the presence or absence of ITF2357 in A375 cells. Western blot analysis of BRAF was performed as reported in Materials and Methods. Representative blots of three independent experiments and densitometric analysis are shown. * *p* < 0.05,** *p* < 0.01, *** *p* < 0.001 with respect to controls; ## *p* < 0.01, ### *p* < 0.001 with respect to ITF2357 treated cells.

**Figure 5 biomedicines-10-01994-f005:**
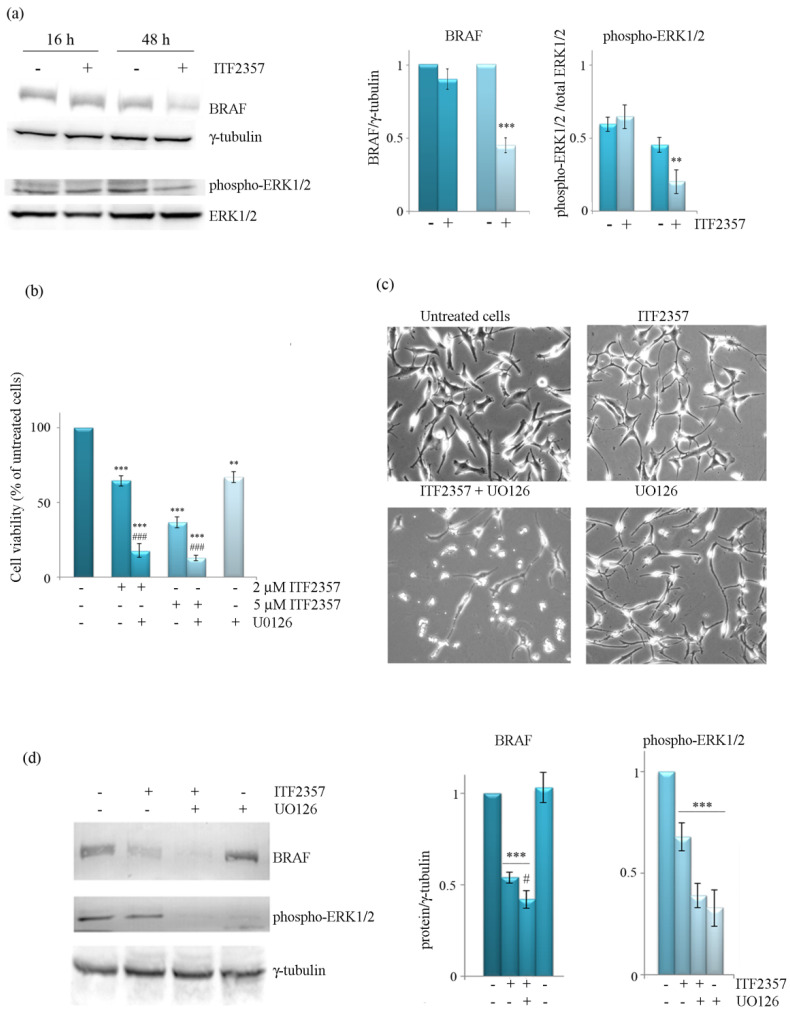
The effects of ITF2357 on phospho-ERK1/2 and potentiation by the MEK inhibitor U0126. (**a**) Western blot analysis of BRAF and phospho-ERK1/2 after 16 and 48 h treatment of SK-MEL-28 cells with 5 μM ITF2357. The effects of ITF2357 in the presence of U0126 on cell viability (**b**) and cell morphology (**c**). Cells were treated with ITF2357 at the indicated concentrations in the absence or presence of U0126 (10 μM) for 48 h. Cell viability was then assessed by MTT assay as reported in Materials and Methods. For morphological analysis, cells were treated for 24 h with 5 μM ITF2357 in the absence or presence of 10 μM U0126. The cells were visualized under light microscope at 200× magnification and the pictures acquired by IM50 Leica Software (Leika Microsystems, Wetzlar, Germany). (**d**) Western blot analysis of BRAF and phospho-ERK1/2 following ITF2357 treatment for 48 h in the absence or presence of U0126. Representative blots of three independent experiments and densitometric analysis are shown. ** *p* < 0.01, *** *p* < 0.001 with respect to untreated cells, # *p* < 0.05, ### *p* < 0.001 with respect to ITF2357 treated cells.

**Figure 6 biomedicines-10-01994-f006:**
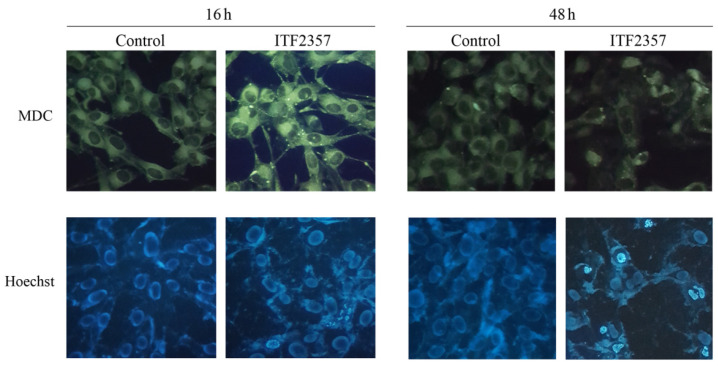
ITF2357 induces autophagic vacuolization and chromatin condensation. SK-MEL-28 cells were incubated for 16 or 48 h in the presence 5 μM ITF2357. At the end of incubation, cells were stained with monodansylcadaverine (MDC), which highlights autophagic vacuoles or Hoechst 333428 and permits the visualization of nuclei. Cells were then visualized under fluorescence microscope Leika equipped with a DAPI filter (Hoechst 333428) or FITC filter (MDC) at magnification of ×400. Micrographs are representative of two different fields from two independent experiments.

**Figure 7 biomedicines-10-01994-f007:**
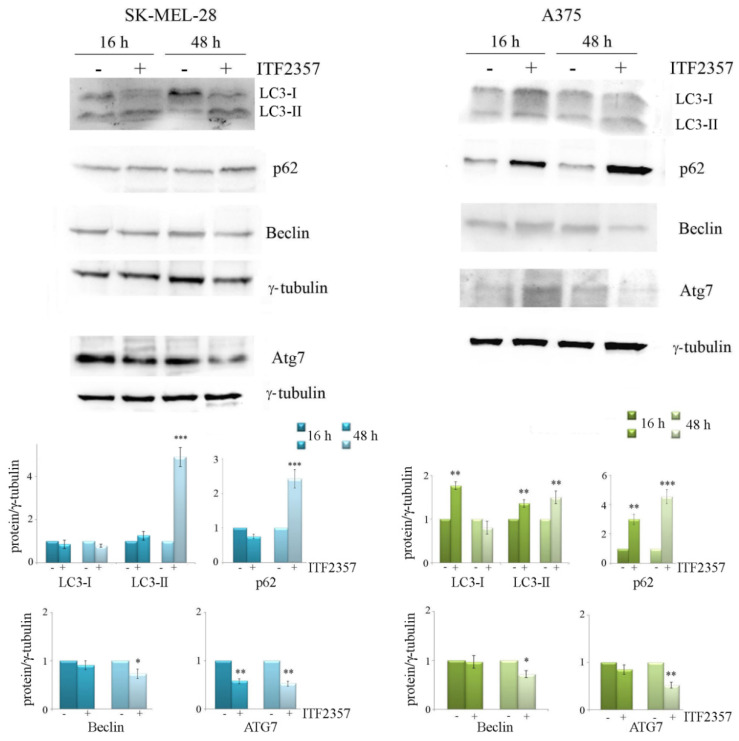
The effects of ITF2357 on autophagic markers. SK-MEL-28 and A375 cells were treated for 16 and 48 h with 5 μM and 2 μM ITF2357, respectively. Western blot analysis of autophagic proteins was performed as reported in Materials and Methods. Representative blots of three independent experiments and densitometric analysis are shown. * *p* < 0.05, ** *p* < 0.01, *** *p* < 0.001 with respect to untreated cells.

**Figure 8 biomedicines-10-01994-f008:**
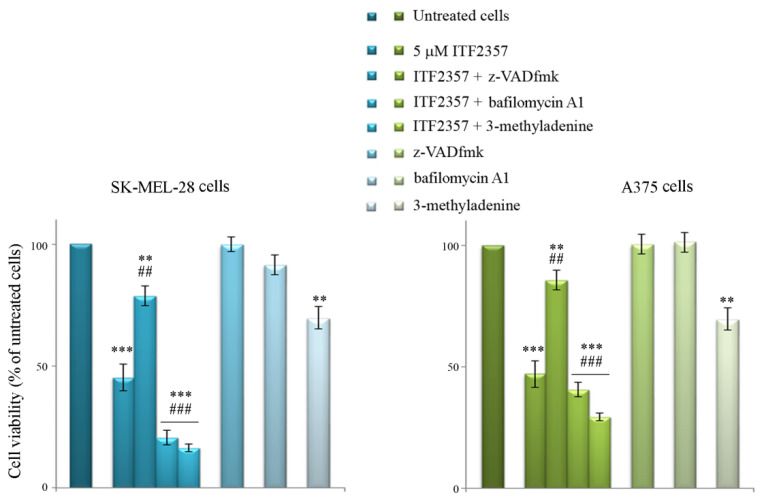
The influence of autophagy inhibitors and pan-caspase inhibitors on the effects of ITF2357. SK-MEL28 and A375 cells were pre-treated for two hours with the autophagy inhibitors bafilomycin A1 (20 nM) or 2.5 mM 3-methyladenine (2.5 mM), then 5 μM ITF2357 was added, and the incubation was protracted for 48 h. Co-treatment ITF2357 with the caspase inhibitor z-VADfmk (80 μM) was maintained for 48 h. MTT assay was performed as indicated in Materials and Methods to evaluate cell viability. The results reported in the histograms are representative of three independent experiments. ** *p* < 0.01, *** *p* < 0.001 with respect to untreated cells; ## *p* < 0.01, ### *p* < 0.001 with respect to ITF2357 treated cells.

**Figure 9 biomedicines-10-01994-f009:**
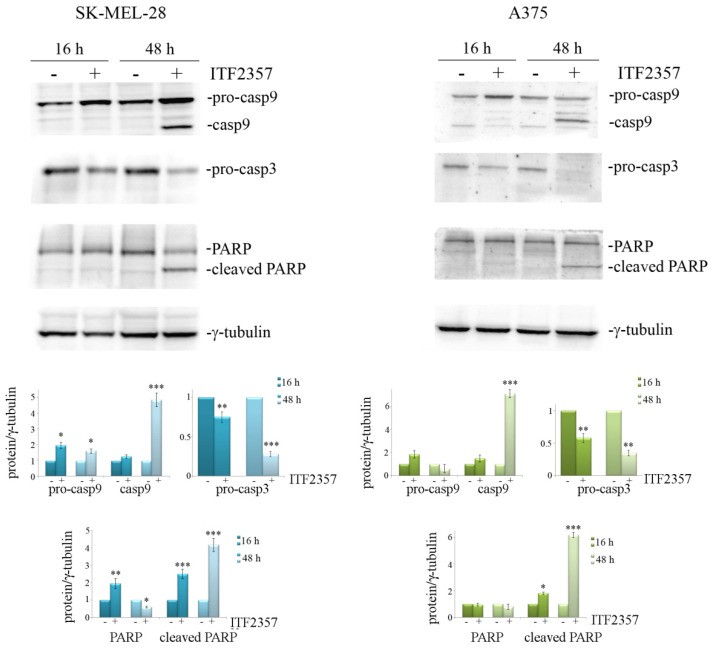
The effects of ITF2357 on apoptotic markers. SK-MEL-28 and A375 cells were treated for 16 and 48 h with 5 μM and 2 μM ITF2357, respectively. Western blot analysis of apoptotic proteins was performed as reported in Materials and Methods. Representative blots of three independent experiments and densitometric analysis are shown. * *p* < 0.05, ** *p* < 0.01, *** *p* < 0.001.

**Table 1 biomedicines-10-01994-t001:** IC50 values of ITF2357 and SAHA in SK-MEL-28 and A375 cells.

	IC50 Value	
	SK-MEL-28 Cells	A375 Cells
ITF2357	4.2 μM	1.7 μM
SAHA	26.9 μM	9.2 μM

## Data Availability

Not applicable.

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
