# Peer review of "The Histone Deacetylase Inhibitor ITF2357 (Givinostat) Targets Oncogenic BRAF in Melanoma Cells and Promotes a Switch from Pro-Survival Autophagy to Apoptosis"

_biomedicines, 2022, doi:10.3390/biomedicines10081994_

Round 1
Reviewer 1 Report
The underlying molecular mechanism is unclear: how does the increase of histone acetylation lead to the decrease of the BRAF protein? The manuscript should be further strengthened by adding some in-depth analysis of the downstream molecular mechanism. Authors should use RNA seq to identify the altered gene expression and the affected pathways.
Figure 4b, the bar chart in the lower panel cannot accurately reflect the result of the upper panel: in fact, the rescue effect of Bortezomib is almost negligible in the A375 cells. Authors may try other proteasome inhibitors such as MG132. They should also examine the BRAF protein half-life using cycloheximide in cell with/without ITF2357 treatment.
Figure 7, the expression of P62 and Beclin shows no difference in SK-MEL-28 cell.
Line 21, MEKK>MEK? Line 19: the writing makes no sense here.
Reviewer 2 Report
The authors of the article studied the effect of the pan HDAC inhibitor ITF2357 (Givinostat) compared to SAHA (Vorinostat) in melanoma cells carrying the BRAF V600E oncogenic mutation. The results show that both ITF2357 and SAHA dose-dependently reduce the viability of BRAF-mutated SK-MEL-28 and A375 melanoma cells. 1) Figure 1 shows the effects of ITF2357 and SAHA on melanoma cell viability. However, the concentration for ITF2357 and SAHA is different, it is not possible to compare the slope of the curve in the same coordinates. Please add the graph of effects of ITF2357 and SAHA on melanoma cell viability with the same concentrations on the x-axis and match them. 2) Why was treatment done with 5 µM (SK-MEL-28) or 2 µM (A375) ITF2357 and not the same concentration in both cases? 3) Figure 3 is also a note. You bring different cell lines with different inhibitors and everywhere the scale is different. It seems to me that it would be more clear to give the same scale of concentration. Explain why different concentrations of inhibitors are used for different cell lines. 4) Authors should describe the results separately and the discussion separately. In this case, the principle of choosing inhibitor concentrations should be more clearly described, as well as the fact that, starting from Figure 4, there is no side-by-side comparison of ITF2357 and SAHA.
Round 2
Reviewer 1 Report
The authors did not fully address my concerns, especially the q1 (underlying molecular mechanism).
Reviewer 2 Report
I have no more comments on the article. I believe that in its present form the article can be recommended for publication.
